# Immune Checkpoint Inhibitor Therapy and Associations with Clonal Hematopoiesis [note 1]

**DOI:** 10.3390/ijms252011049

**Published:** 2024-10-15

**Authors:** Abhay Singh, Nuria Mencia Trinchant, Rahul Mishra, Kirti Arora, Smit Mehta, Teodora Kuzmanovic, Maedeh Zokaei Nikoo, Inderpreet Singh, Amanda C. Przespolewski, Mahesh Swaminathan, Marc S. Ernstoff, Grace K. Dy, Lunbiao Yan, Eti Sinha, Shruti Sharma, Duane C. Hassane, Elizabeth A. Griffiths, Eunice Wang, Monica L. Guzman, Swapna Thota

**Affiliations:** 1Leukemia and Myeloid Disorders Program, Cleveland Clinic, Cleveland, OH 44106, USA; 2Division of Medicine, Weill Cornell Medical College, New York, NY 10065, USA; 3Department of Internal Medicine, Anne Arundel Medical Center, Annapolis, MD 21401, USA; 4Department of Medicine, Cleveland Clinic Akron General Hospital, Akron, OH 44307, USA; 5University Hospitals, Case Western Reserve University, Cleveland, OH 44106, USA; 6Upstate Community Hospital, SUNY Upstate Medical University, Syracuse, NY 13210, USA; 7Department of Medicine, Roswell Park Comprehensive Cancer Center, Buffalo, NY 14203, USAgrace.dy@roswellpark.org (G.K.D.);; 8Tempus Labs, Inc., Chicago, IL 60654, USA; 9Department of Medicine, The University of Tennessee Health Science Center, Memphis, TN 38103, USA

**Keywords:** immune checkpoint blockade, immune checkpoint inhibitor, clonal hematopoiesis of indeterminate potential, *DNMT3A*, TET2, non-small cell lung cancer, melanoma

## Abstract

Cancer cohorts are now known to be associated with increased rates of clonal hematopoiesis (CH). We sort to characterize the hematopoietic compartment of patients with melanoma and non-small cell lung cancer (NSCLC) given our recent population level analysis reporting evolving rates of secondary leukemias. The advent of immune checkpoint blockade (ICB) has dramatically changed our understanding of cancer biology and has altered the standards of care for patients. However, the impact of ICB on hematopoietic myeloid clonal expansion remains to be determined. We studied if exposure to ICB therapy affects hematopoietic clonal architecture and if their evolution contributed to altered hematopoiesis. Blood samples from patients with melanoma and NSCLC (*n* = 142) demonstrated a high prevalence of CH. Serial samples (or post ICB exposure samples; *n* = 25) were evaluated in melanoma and NSCLC patients. Error-corrected sequencing of a targeted panel of genes recurrently mutated in CH was performed on peripheral blood genomic DNA. In serial sample analysis, we observed that mutations in *DNMT3A* and *TET2* increased in size with longer ICB exposures in the melanoma cohort. We also noted that patients with larger size *DNMT3A* mutations with further post ICB clone size expansion had longer durations of ICB exposure. All serial samples in this cohort showed a statistically significant change in VAF from baseline. In the serial sample analysis of NSCLC patients, we observed similar epigenetic expansion, although not statistically significant. Our study generates a hypothesis for two important questions: (a) Can *DNMT3A* or *TET2* CH serve as predictors of a response to ICB therapy and serve as a novel biomarker of response to ICB therapy? (b) As ICB-exposed patients continue to live longer, the myeloid clonal expansion may portend an increased risk for subsequent myeloid malignancy development. Until now, the selective pressure of ICB/T-cell activating therapies on hematopoietic stem cells were less known and we report preliminary evidence of clonal expansion in epigenetic modifier genes (also referred to as inflammatory CH genes).

## 1. Introduction

Selective expansion of clonal hematopoiesis (CH) mutations in DNA damage response pathway genes following chemotherapy, radiation therapy and radionuclide therapy exposure is well established among cancer survivors [1,2,3,4]. Analyzing exposures that lead to a selective growth advantage of CH is an active investigational focus [5,6]. Cancer survivors frequently harbor clonal hematopoiesis of indeterminate potential (CHIP or CH) mutations [7,8,9,10]. The number of cancer survivors living in the United States continues to increase each year as a result of the growth and aging of our population, as well as an increase in survival due to changes in early detection and treatment advances [11]. Immune checkpoint blockade (ICB) therapy has revolutionized outcomes among patients with various primary malignancies [12]. The selective pressures of T-cell activating therapies on the hematopoietic compartment are unclear. We previously reported a change in the landscape of secondary myeloid malignancies in a large US population database since the dawn of ICB therapy [13]. In the current study, we aimed to evaluate the impact of exposure to ICB therapy on hematopoietic clones (or CH) in patients with non-small cell lung cancer (NSCLC) and melanoma (MEL). In addition, some data suggest that CH mutations may have a role in preventing T-cell exhaustion and thereby, enhance antitumor activity [14,15]. Therefore, in a subset of patients, we studied the association between hematopoietic clones and duration of response to ICB, hypothesizing specific CH clones as potential novel biomarkers of response.

## 2. Results

We identified 167 samples from patients (*n* = 142) with a diagnosis of MEL (*n* = 33) and NSCLC (*n* = 109), treated with ICB at our institution. Forty-eight percent in MEL and 57% in NSCLC cohorts were females. Thirty percent of MEL and 85% of NSCLC cohorts had a history of smoking. The median ages of the MEL and the NSCLC cohorts were 63 and 68 years, respectively. Fifty-six percent and 55% of the MEL and the NSCLC cohorts, respectively, had metastatic disease (Table 1). Our cohort had minimal baseline chemotherapy and/or ICB exposure. Most baseline samples were collected prior to chemotherapy (CTX) or ICB exposure. Ten percent of the MEL cohort had CTX or targeted therapy exposure prior to initial sample collection and ICB treatment initiation. Forty-eight percent of the NSCLC cohort had prior CTX (Table 1).

### 2.1. CH in NSCLC and MEL

We identified CH at VAF > 1% in 22.6% (7 of 31 evaluable patients) and 37.6% (41/109) of the baseline MEL and NSCLC samples, respectively (Figure 1A,B). When using a VAF cut-off of >2%, CH incidence was 16.1% (5/31) in MEL and 24.7% (27/109) in patients with NSCLC. Ten percent and 19.5% of patients in the MEL and NSCLC samples, respectively, had mutations in more than one CH gene. In both groups, *DNMT3A* (*n* = 27), *TET2* (*n* = 7) and *ASXL1* (*n* = 6) were the most common genes found to be mutated. The mean VAF in the MEL cohort was 4.9% (1.01–19.1%) and 2.27% (1.01–26.4%) in the NSCLC cohort. We defined cohorts with CH at VAF > 2% compared to no CH cohorts in MEL and NSCLC patients. Patients with CH were older in age compared to those without CH, 70 vs. 62 years (*p* = 0.12) in the MEL cohort and 74 vs. 66 years (*p* = 0.00012) in the NSCLC cohort. CH patients had normal hematological parameters with the exception of increased red cell distribution width (RDW; a recently described progression predicting variable [16], RDW of 15.2 vs. 13.2 in MEL (*p* = 0.0071). Gender and other blood counts were assessed as a factor of CH, and no significant associations were noted.

### 2.2. Interaction of CH and ICB

Primary tumor responses in this cohort were defined as durable (receipt of ≥12 ICB cycles). All *DNMT3A* and *TET2* mutations expanded over time after ICB exposure (Figure 2a). In patients with MEL, CH with higher VAF was associated with longer duration of ICB therapy (as an example in Figure 2b; plot titled ‘Immunotherapy_cycles’; PT-00298654 and PT-00306840 had VAFs 10% or more and received 25 or more ICB cycles). MEL patients with a *DNMT3A* mutation (*n* = 5) had a trend towards durable ICB responses (≥12 ICB cycles), i.e., received a higher median number of ICB cycles (21 cycles, range: 10–40) compared to non-*DNMT3A* CH mutant patients (7, range: 1–13; *p* = 0.21). Despite the significantly older age of the *DNMT3A* CH+ cohort (67.8 years; 52–88) versus the CH negative cohort (mean age 60.7; 33–77), both groups had an equivalent duration of ICB exposure (Appendix A). Additionally, as discussed above, patients with larger *DNMT3A* clones tended to receive a higher number of ICB cycles (Figure 2b). In the serial sample analysis, we observed that mutations in *DNMT3A* and *TET2* increased in size with longer ICB exposures in the MEL cohort (Figure 2a and Appendix A). Three patients in the MEL cohort received >15 ICB cycles. Another patient (PT-00306840) with the most notable *DNMT3A* expansion received 40 ICB cycles (Figure 2a and Appendix A). In the serial sample analysis for the NSCLC cohort, we observed that patients with ≥3 months of ICB exposure demonstrated a decrease in clone size in gene mutations such as *SRCAP*, *STK11* and *TPM1* (Appendix A). Again, exceptions were *DNMT3A* and *TET2*, which showed stability or increase in size with longer ICB exposure. The characteristics of patients whose serial samples were available are shown in Appendix A.

## 3. Discussion

High CH prevalence was noted in solid malignancies and is associated with poor survival [8,17]. Receipt of CTX and RT has been associated with the emergence or propagation of DNA damage repair (DDR) pathway gene mutations [4,7,8]. Mutations in epigenetic modifiers, i.e., *DNMT3A* and *TET2* are not frequently affected by exposure to such therapy [7,18]. Large studies published thus far did include a subset that received ICB and targeted therapy. Associations of specific clonal dynamics for the patients on ICB have not been rigorously studied by serial sample analysis. In an attempt to answer this, a prior study investigated the impact of ICB on CH by analyzing blood samples from 91 patients with cutaneous melanoma and basal cell carcinoma before and after ICB treatment [17]. The results indicated that ICB treatment did not significantly affect the prevalence, size, or mutational landscape of CH and they concluded that ICB did not drive clonal evolution. This study, however, focused on clonal expansions in DDR pathway genes, such as *TP53* and *PPM1D* [17]. Our study performed sequential genomic sequencing and noted that VAFs either increased or decreased based on driver gene mutation while on ICB. We particularly noted expansions within genes involved in inflammatory pathways, i.e., *DNMT3A* and *TET2*. A plausible explanation for this discordance possibly stems from an insufficient time period to observe a meaningful clonal expansion in the prior study as well as the study question involving multiple CH mutations, as opposed to our serial cohort enriched with majorly inflammation-related mutations of *DNMT3A* and *TET2*.

*DNMT3A* clones originate at an early age and gradually expand at a steady rate [19]. The expected *DNMT3A* clonal expansion rate varies considerably with different exposures and mediators [19]. We demonstrate in our MEL cohort a significant *DNMT3A* clonal expansion among serial samples and a *TET2* clonal expansion (Appendix A). An increase in VAF of CH-clones (predominantly *DNMT3A*) was associated with a longer duration of ICB therapy exposure. A certain limitation of our serial sample analysis is our inability to procure several serial samples on *DNMT3A/TET2* mutant patients in the NSCLC cohort. However, our study generates two essential and provocative hypotheses that warrant further investigation: (a) ICB and inflammatory-CH (iCH; *DNMT3A/TET2* related CH) share a bidirectional relationship where iCH may enhance the response to ICB and the response to ICB further drives iCH (as these mutant clones thrive under inflammation); (b) ICB-mediated iCH expansion may lead to a novel therapy-related myeloid neoplasm as patients on ICB therapies continue to live longer. To test the former, iCH as a potential novel biomarker of response, larger studies with numerous sequential samples are needed whereas, for the latter, several years of follow-up would be prudent. Overall, we noted that CH in epigenetic modifier genes (*DNMT3A*/*TET2*) was associated with a durable response to ICB therapy in ICB-exposed patients. The presence of *DNMT3A*/*TET2*-CH and their increasing clonal burden correlated with longer ICB exposure. The precise mechanisms behind these findings have not been clarified. One compelling hypothesis is that improved response to ICB and thus induction of inflammatory milieu provides selection pressure for fitter clones that thrive under inflammation.

It is important to note that, even in cases of ICB-sensitive disease, prolonged ICB therapy may not be feasible for some patients due to intolerable side effects, particularly those that are immune-mediated. A higher baseline iCH may not be able to predict ICB duration and response correctly in such cases. The inability to capture such cases is a limitation of our study. Another limitation is the lack of an external control group in our study. Ideally, a control group of patients with metastatic melanoma or NSCLC who had not received immune checkpoint inhibitors (ICBs) would have been included. However, this is challenging since ICBs are the current standard of care, especially for melanoma, and historical controls are hard to establish due to inconsistent sample availability in our biorepository or lack of comparable methods for assessing clonal hematopoiesis (CH), particularly when compared to chemotherapy. Given these limitations, we focused our conclusions on pre- and post-treatment changes in variant allele frequency (VAF), consistent with prior CH studies. Lastly, most baseline samples were collected prior to CTX or ICB exposure, however, ten percent of the MEL cohort had CTX or targeted therapy exposure prior to initial sample collection and ICB treatment initiation. Hence, part of the clone expansion observed maybe due to a previous insult in such patients, another limitation of this study.

Nonetheless, our findings and resultant hypothesis are well-supported by increasing evidence that CH modulates immune and inflammatory pathways in both clinical and in vivo settings [20]. Findings that support this observation include recent reports of *DNMT3A* CH in hematopoietic stem cell (HSC) grafts, which were shown to produce better outcomes in HSC transplant patients who did not receive post-transplant graft vs. host disease prevention with cyclophosphamide (PTCy) [15]. This observation suggests the role of T-cells and resultant inflammation as a permissive condition for the expansion of *DNMT3A* clones or vice versa. Whether heightened immune activation of T-cells in the setting of CH drives longer and better responses to ICB therapy, represents an important yet-to-be-answered question. Additionally, previously, *DNMT3A* CH evolving during effective ICB appeared to serve as a predictor of improved survival in NSCLC patients [21]. The precise mechanisms need clarification in future larger studies of sequential genomic analysis; however, our study generates a hypothesis that induction of pro-inflammatory milieu under ICB stress provides selection pressure for more fit HSC clones that thrive under inflammation or are less inflammation sensitive.

Preclinically, macrophages carrying the *DNMT3A* mutation enhance the inflammatory response in mouse models [22]. At the same time, pro-inflammatory states reciprocally were shown to enhance *DNMT3A* clonal expansion in pre-clinical and clinical models [22]. The loss of function *TET2* gene allows a switch of immunosuppressive tumor-associated macrophages to proinflammatory ones. It was also recently noted that deletion of *DNMT3A* in T-cells can prevent T-cell exhaustion and enhance anti-tumor activity [16]. In another study, a zebrafish model of CH suggested that the clonal fitness of mutant clones is driven by enhanced resistance to inflammatory signals from their *mutant* mature cell progeny. Using an approach called TWISTR (tissue editing with inducible stem cell tagging via recombination), authors identified a survival pathway within the mutant HSPCs involving inflammatory modulators. On a clinical level, in addition to the *DNMT3A*-PTCy discussion above, studies also suggest that deleting *DNMT3A* in CAR-T-cells prevents exhaustion and enhances antitumor activity [23]. The patients with prolonged cytopenia after CAR-T-cell therapy had bone marrow infiltration by interferon-gamma-producing CD8+ T-cells. In a remote way, this suggests that heightened T-cell activity, such as might occur with ICB treatment, could affect the bone marrow in a way that could favor and select certain CH clones [24]. These pre-clinical and clinical observations, including our study findings of *DNMT3A* expansion while on ICB therapy, provide ground for designing functional assays that link epigenetic modification from mutant *DNMT3A* to T-cell expansion; thereby setting grounds for evaluating iCH as a novel biomarker of ICB response.

## 4. Materials and Methods

### 4.1. Study Methods

We accessed and analyzed blood samples from the biorepository of Roswell Park Cancer Center (RPCCC) for patients with a diagnosis of NSCLC or MEL and treated with ICB. Patients were screened and identified through a clinical database as well as the biorepository for receipt of ICB therapy. Stored samples (*n* = 142); with a diagnosis of MEL (*n* = 33) and NSCLC (*n* = 109) treated with ICB at RPCCC were identified and analyzed. Additionally, serial prospective peripheral blood samples from subset of patients (NSCLC, *n* = 14, and MEL, *n* = 11) were obtained after informed consent. Clinical and laboratory variables were collected for the patients through retrospective chart review. Peripheral blood counts were determined using complete blood counts obtained at the time of sample collection (±30 days). Chemotherapy or radiotherapy exposure were based on receipt of any systemic therapy that patients had received at or after initial blood/DNA sampling. No patients in our analysis had another active hematologic malignancy or a precursor state such as monoclonal gammopathy of undetermined significance or monoclonal B-cell lymphocytosis at the time of peripheral blood sequencing. The study was approved by the Institutional Review Board (approval number BDR-113119).

### 4.2. Mutational Analysis

Genomic DNA from mononuclear cells or buffy coat was extracted. Illumina sequencing adapters containing unique dual sample indices (UDIs) were ligated onto fragments according to the Kapa Biosystems HyperPrep protocol. Targeted enrichment using a custom pool of biotinylated baits directed to 93 genes involved in clonal hematopoiesis, cancer, and cardiovascular disease (CVD) was performed according to the standard Twist Biosciences capture protocol. Sequencing was performed to 1800× mean overall depth of coverage using NovaSeq6000 (Illumina inc., San Diego, CA, USA) using 150 × 150 S4 chemistry. Clonal hematopoiesis (CH) detection followed the strategy in previous studies [25] for quality and filtration of artifacts. Gencore v0.13.0 [26] and fastp v0.20.1 [27] were used to trim adaptor sequences and remove duplicates. After alignment with BWA MEM v0.7.17 [28], VarDict-Java v1.7.0 [29], was used to call variants on reads. Bcftools v1.10.2 [30], SnpEff v4.3 [31], and Vcflib v1.0.1 [32], were used to filter out reads for mapping quality, depth, and strand bias. Variant-Effect-Predictor (VEP) v100.3 [33], was used to annotate variants. SNP were stringently filtered out when reported as >0.25% in population allele frequency databases. Variants with VAF ≥ 1% are being reported. The sensitivity of the test is estimated at >95% for mutations present at VAF > 1% (corresponding to 1/50 mutated cells).

### 4.3. Quantification and Statistical Analysis

Analysis was conducted in R environment (https://www.R-project.org; R: A language and environment for statistical computing. R Foundation for Statistical Computing, Vienna, Austria, accessed on 1 July 2021). Figures were produced using the package ggplot2 also in R v3.3.3 (Wickham, NZ, 2016). Statistical significance was defined as *p* < 0.05. Statistical tests were performed using R software. As per the procedure, we used a linear model for VAF changes. For each mutation in each individual with sequential sequencing data available, we modeled the growth rate of the mutation between the two time points according to the following formula: a = log (V/V0)/(T − T0) Where T and T0 indicate the age of the individual (in days) at the two measurement time points and V and V0 correspond to the VAF at T and T0, respectively. We also classified mutations as having increased, decreased, or remained constant during the follow-up period based on a binomial test comparing the two VAFs. Generalized estimating equations were used to test for an association between exposure to ICB and CH growth rate adjusting for age, gender, and smoking status accounting for correlation between the growth rate of mutations in the same person. A paired *t*-test was used to test for significance in the difference between growth rates of mutations within individuals who received ICB. List of genes in CH panel (*n* = 93) is in Appendix A.

## 5. Conclusions

To summarize, our study raises two important questions to be addressed: (a) Can *DNMT3A* or *TET2* CH serve as a predictor of response to ICB therapy? (b) As ICB-exposed patients continue to live longer, the myeloid clonal expansion may portend an increased risk for subsequent myeloid malignancy development. ICB-induced CH may predispose patients to a novel treatment-related hematopoietic neoplasm, which will need to be defined with longer follow-ups and larger studies. Additionally, our study, in alignment with prior studies, demonstrates *DNMT3A* and *TET2* to be the common genes to be mutated among cancer cohorts. Despite the small size of the cohort, owing to optimal latency time between serial samples, we noted *DNMT3A* clonal expansion while on ICB therapy. As a future direction, larger studies with sequential samples are needed to confirm this observation as well as functional assays to study the epigenetic advantage for T-cell expansion with mutant *DNMT3A* or *TET2*.

## Figures and Tables

**Figure 1 ijms-25-11049-f001:**
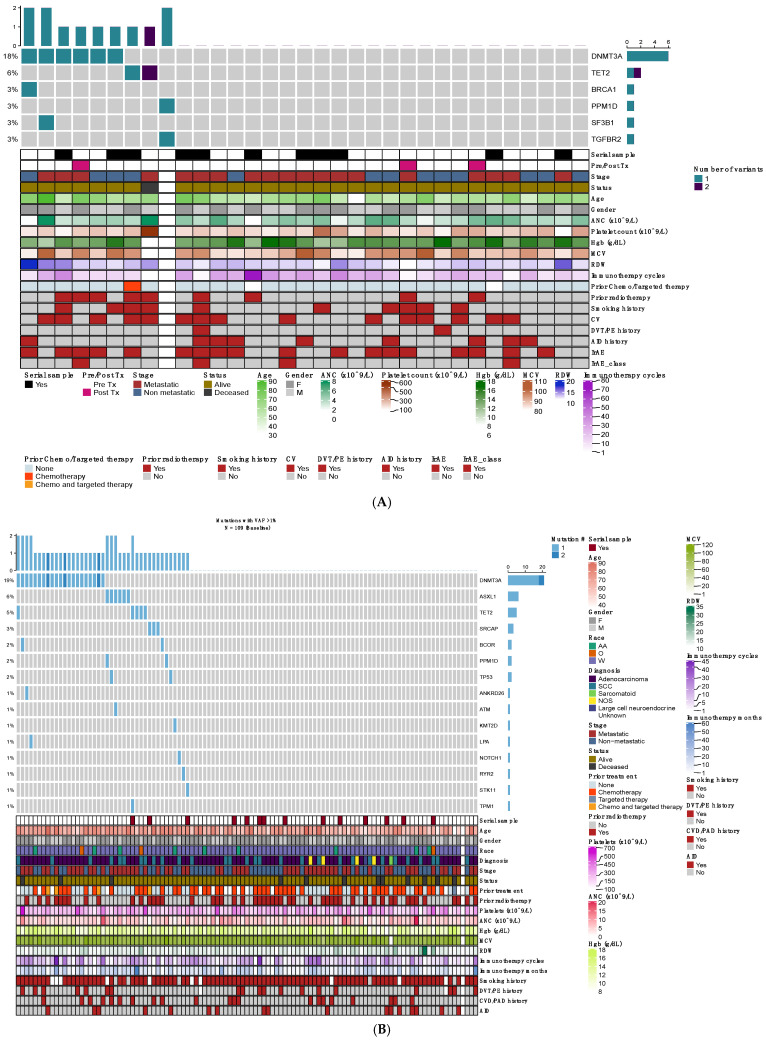
(**A**) Description of clinical, hematological, and genetic characteristics of patients with melanoma. (**B**) Description of clinical, hematological, and genetic characteristics of patients with non-small cell lung cancer. ANC (absolute neutrophil count); Hgb (hemoglobin); MCV (mean corpuscular volume, fL); RDW (red cell distribution width, %); CV (cardiovascular disease); DVT/PE (Deep vein thrombosis/Pulmonary embolism); AID (Autoimmune disease); IrAE (Immune-related adverse events). Race (AA: African American; O: other; W: White), SCC: Squamous cell cancer; NOS (Not otherwise specified); CVD/PAD (cardiovascular disease/Peripheral arterial disease).

**Figure 2 ijms-25-11049-f002:**
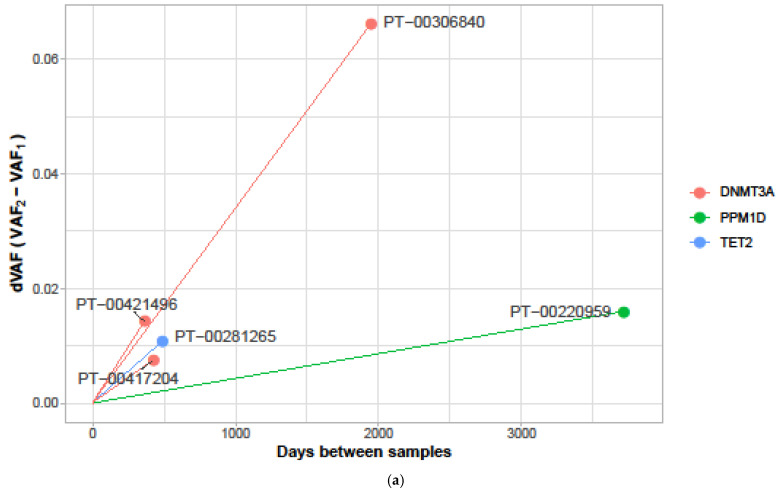
(**a**) Change in clonal architecture in serial blood samples after exposure to immune checkpoint inhibitor (ICB) in five patients with melanoma. Spider plot depicts difference in variant allele frequencies (VAF) (y-axis) in blood samples pre- and post-ICB therapy ICB (Immune checkpoint inhibitor) plotted against time between pre- and post-sample collection (x-axis); Pre-ICB sample collection at day 0. (**b**) Age, hematological and treatment characteristics in patients with melanoma and clonal hematopoiesis mutations, at baseline. ANC (absolute neutrophil count, 10^3^/µL); Hgb (hemoglobin, gm/dL); MCV (mean corpuscular volume, fL); RDW (red cell distribution width, %); VAF (Variant allele frequency).

**Table 1 ijms-25-11049-t001:** Characteristics of patients with non-small cell lung cancer (NSCLC) and Melanoma (MEL).

	NSCLC (*n* = 109)	MEL (*n* = 33 *)
Age (years)	67.6	62.5
Range	33–89	33–88
Gender		
Female (%)	62 (56.8%)	16 (48.5%)
Stage		
Metastatic (%)	60 (55%)	19 (57.5%)
Smoking		
Yes (%)	93 (85.3%)	10 (30.3%)
Vascular disease		
Yes (%)	21 (19.3%)	15 (45.5%)
Deep vein thrombosis/Pulmonary embolism		
Yes (%)	19 (17.4%)	2 (6%)
Exposures prior to baseline		
No exposure to chemo/targeted therapy	38 (34.9.9%)	29 (88%)
Radiotherapy exposure	51 (46.7%)	9 (27%)
Targeted and Chemotherapy exposure	4 (4%)	0 (0%)
Treatment received		
Immune checkpoint blockade (ICB)	82 (75%)	29 (88%)
Number of ICB cycles (median)	13.6 (2–44)	22 (1–71)
Blood counts		
ANC (10^3^/µL)	5.7 (1–20)	4.4 (1.9–7.6)
Platelets (10^3^/µL)	306 (127–617)	238 (134–531)
Hemoglobin (gm/dL)	12.7 (8.2–16.8)	14 (8.4–16.6)
MCV (fL)	90.4 (3.5–107.8)	91.6 (82–102)
RDW (%)	14.5 (11.9–32.2)	13.5 (12–17.7)

* 1 patient with missing clinical data. ANC: Absolute neutrophil count; MCV: Mean corpuscular volume; RDW: Red cell distribution width.

## Data Availability

The data presented in this study are available on request from the corresponding author. The data are not publicly available due to privacy or ethical restrictions.

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
