# Peer review of "Immune Checkpoint Inhibitor Therapy and Associations with Clonal Hematopoiesis†"

_ijms, 2024, doi:10.3390/ijms252011049_

Round 1

Reviewer 1 Report

Comments and Suggestions for Authors

                The authors have provided an extensive amount of data concerning the incidence of CH, and associated measures of bone marrow output, in patients with metastatic melanoma or NSCLC who are treated with ICBs. I find no fault with how the work was done, and the conclusions seem logical. However, I have questions whose answer I may have missed finding in the manuscript, or which need to be provided by the authors.

1) What is the control group for the study? I found myself confused by this question, and I think that the authors need to explain clearly their approach to this issue. To determine whether ICB has an effect on CH, the most ideal control group would be patients with metastatic melanoma or NSCLC who have not been treated with ICBs, but having such a group would be problematic. At least for melanoma, ICBs are the current standard of care, so the only practical option would be historical controls, but the samples may not be available, or CH was not previous determined in a comparable manner. I am not certain, but I think that since everyone in the study received ICBs, the authors have used correlation with duration of ICB treatment as the “control” (i.e., basis) by which to determine whether ICB has an effect on CH. However, this is problematic because duration of ICB treatment is subject to multiple factors that could be confounding. Patients may stop ICB treatment early due to intolerable autoimmune side effects or to lack of tumor control, both of which could involve different physiological states with potential effects on CH. Therefore, patients who are on ICB longer are also less likely to have intolerable autoimmune side effects or uncontrolled tumor growth. The authors should acknowledge these limitations of their study, or provide historical controls if available.

2) Is it justified to suggest that VAF of DNMT3A or TET2 could serve as a biomarker predictive of response to ICB therapy? It appears that the authors regard duration of ICB treatment as an indicator of successful tumor control. While it is true that patients are unlikely to remain on ICB if their tumors are not responding, the frequent incidence of intolerable autoimmune side effects makes it improper to assume that patients not on ICB had uncontrolled tumor. Therefore, it seems invalid to assume that since patients with longer duration of ICB treatment had higher VAF for DNMT3A or TET2, that these could serve as a biomarker predictive of response to ICB therapy. The authors should defend their interpretation.

3) What is the mechanism by which ICB treatment might accelerate the progression of CH, or at least an increase in the VAF of DNMT3A or TET2? I do not expect the authors to be able to answer this question definitely, and indeed they have provided some possible mechanisms. They may find PMID: 37586321 to be relevant, in which it was shown that patients with prolonged cytopenia after CAR-T cell therapy had bone marrow infiltration by IFN-gamma producing CD8+ T cells. In a remote way, this suggests that heightened T-cell activity, such as might occur with ICB treatment, could affect the bone marrow in a way that could favor certain CH clones.

Reviewer 2 Report

Comments and Suggestions for Authors

In this paper the authors analyse the CH landscape in a cohort of cancer patients undergoing ICB and investigate the potential interplay between CH and the treatment. The manuscript is interesting, and the availability of serial samples enriches the study. Following my comments.

Abstract: The authors state that the VAF comparison was performed with Fisher test, whereas in methods section they describe a formula to assess this variable. I would delete this statement from the abstract or describe the methods more in deep.

Methods: Please, for consistency, explain MGUS and MBL acronyms.

Supplemental table 2 is referred as supplemental table 1 in the text.

Results: “CH at 2% VAF cutoff correlates with clinical outcomes” (Line 136): Is this statement a sum of the findings or a background information? In this second case studies should be cited.

Results: Was RDW significantly different between CH and no CH? A p value should be reported as for the other variables.

An other table with the characteristics of patients whose serial samples were available would enrich the paper.

Figures are divided into panels (A/B) even if they are not strictly related, especially in supplemental. I would divide figures (e.g. supplemental 2 figures). Furthermore, there is supplemental table 1 A but not 1B. What is the purpose?

The second result paragraph is quite confounding, especially where the authors talk about serial samples and baseline ones. I would split the two things or even discuss them in two separate paragraphs.

I would include aa limitation of the analysis the fact that part of the patients previously received chemo/radiotherapy and thus, part of the clone expansion observed maybe due to a previous insult.

Discussion: The perspective of therapy-related MN after ICB is a relevant point raised by the authors and maybe deserves more space. Were TP53 clones identified? Did new clones/mutations emerge at the second time point? CH-drivers have different role/trajectory in t-MN development (PMID:39031199). Do the authors think that DNMT3A and TET2 VAF changes over time may have the same clinical implications?

Reviewer 3 Report

Comments and Suggestions for Authors

This manuscript explores the impact of immune checkpoint blockade (ICB) therapy on hematopoietic clones (CH) in patients with non-small cell lung cancer (NSCLC) and melanoma (MEL). It reveals an association between CH and the duration of response to ICB. Notably, the findings show that mutations in DNMT3A and TET2 expand in size with longer ICB exposure. Whether DNMT3A or TET2 CH can serve as predictors or biomarkers of response to ICB therapy requires further investigation. The manuscript can be improved by addressing the following concerns.

Investigating the correlation between CH and ICB response is crucial. The authors used parameters such as the number of days between samples and immunotherapy cycles, but how these correlate with ICB response is not clearly explained in the background.

In Fig. 2a, the correlation between days between samples and dVAF is presented. Does the baseline VAF start at zero? It is important to show both baseline and post-ICB therapy VAFs for a clearer understanding.

Additionally, abbreviations should be defined the first time they are mentioned, such as VAF on line 29.
